# Selinexor’s Immunomodulatory Impact in Advancing Multiple Myeloma Treatment

**DOI:** 10.3390/cells14060430

**Published:** 2025-03-13

**Authors:** Kereshmeh Tasbihi, Heiko Bruns

**Affiliations:** Department of Medicine 5—Hematology and Oncology, University Hospital Erlangen, 91054 Erlangen, Germany; kereshmeh.tasbihi@uk-erlangen.de

**Keywords:** Selinexor, XPOVIO, XPO1, Exportin-1, nuclear export, multiple myeloma, tumor microenvironment, immunotherapy

## Abstract

Despite the major advancements in the repertoire for multiple myeloma (MM) treatment, this disease remains a chronically progressive plasma cell malignancy. Drug resistance and high relapse rates complicate the extended treatment strategies. However, the tumor microenvironment (TME) in MM is decisive for the success of a therapy or relapse. Aiming to improve the outcome of relapsed and refractory MM patients, Selinexor has entered the drug arsenal of myeloma therapy through the implementation of a novel therapeutic approach by selectively inhibiting the nuclear export receptor Exportin-1 (XPO1). Selinexor leads to the inactivation of cancer-related proteins and induces apoptosis by disrupting the nucleocytoplasmic flow in myeloma cells. While this drug is selectively cytotoxic to neoplastic cells, Selinexor’s immunomodulatory impact on the TME is currently being investigated. The aim of this review was to elucidate Selinexor’s capacity to influence the cell interaction network of the TME from an immunological perspective. Deciphering the complex interplay of highly plastic immune cells provides a contribution to the molecular–biological exploration of disease initiation and progression in MM. Unraveling the novel therapeutic targets of the immunological TME and evaluating the advanced immunotherapeutic regimens implementing Selinexor will shape the future directions of immune-oncotherapy in MM.

## 1. Introduction

Multiple myeloma (MM) belongs to the group of B-cell non-Hodgkin lymphoma and is a heterogenous and chronically progressive malignant plasma cell dyscrasia. A characteristic of this disease is abnormal monoclonal plasma cells in the bone marrow (BM)-producing monoclonal immunoglobulins, i.e., paraproteins, which can be quantified in blood and urine. The diagnosis of MM requires the evidence of ≥10% clonal plasma cells upon BM examination or a biopsy-proven plasmacytoma as well as the presence of at least one of the myeloma-defining events. The latter consists of established CRAB criteria (hypercalcemia, renal failure, anemia, or lytic bone lesions) and three specific biomarkers (clonal plasma cells in the BM ≥ 60%, serum free light chain (FLC) ratio ≥ 100 (provided involved FLC level ≥ 100 mg/L) and more than one focal lesion seen by magnetic resonance imaging) [1]. Monoclonal gammopathy of undetermined significance (MGUS) represents the preliminary stage of MM and is defined as serum monoclonal protein level < 30 g/L, clonal plasma cell population in the BM < 10%, and absence of CRAB criteria [2,3,4]. Smoldering MM (SMM) is the precursor of MM and is distinguished from MGUS regarding the serum level of monoclonal protein (≥30 g/L) and the proportion of clonal plasma cells in the BM (≥10%). In the majority of SMM cases, patients are managed by active surveillance; however, in high-risk SMM, specific treatment is also under investigation [5,6,7]. The dissemination of clonal plasma cells into the peripheral blood following previously diagnosed MM represents secondary plasma cell leukemia [8]. Molecular models of disease progression from myeloma precursor states to MM include, inter alia, influences from the tumor microenvironment (TME) and pathogenic triggers, such as genomic aspects [9,10] (Figure 1). The clinical presentation of MM is variable and ranges from asymptomatic states to hematopoietic insufficiency, renal injury, and osteolytic destruction [11]. Symptomatic patients suffer from bone pain and pathologic fractures, anemia, weight loss, hypercalcemia, and secondary immune deficiency, as well as light chain nephropathy [12,13,14,15,16,17]. Detailed conclusions have not yet been reached about the underlying pathogenic triggers, which are probably multifactorial in most patients. Genetic abnormalities and predisposition, epigenetic modification, and dysregulated cellular pathways, as well as clonal heterogeneity, play a remarkable role in the disease manifestation [18].

The therapeutic landscape of MM has significantly evolved in the last two decades. First-line treatment regimens of MM include monoclonal antibodies against cluster of differentiation (CD) 38 (daratumumab, isatuximab), immunomodulatory drugs (IMiDs) (thalidomide, lenalidomide, pomalidomide), proteasome inhibitors (PIs) (bortezomib, carfilzomib, ixazomib), and corticosteroids, in addition to high-dose chemotherapy and autologous stem cell transplantation depending on the patient’s fitness [19]. The treatment of refractory or relapsed cases can be extended by chimeric antigen receptor (CAR) T cell therapy (ciltacabtagene autoleucel, idecabtagene vicleucel) [20,21,22] as well as B-cell maturation antigen (BCMA)-directed (elranatamab, teclistamab) and G protein-coupled receptor class C group 5 member D (GPRC5D)-directed (talquetamab) bispecific antibodies [23,24,25]. Further advanced therapeutic options are offered by an immunostimulatory antibody against signaling lymphocyte activation molecule F7 (SLAMF7) (elotuzumab) [26], as well as Selinexor, the selective inhibitor of nuclear export (SINE) compound [27].

Due to therapeutic innovations, the 5-year overall survival has doubled over the past decade to approximately 54% [28]. However, the incidence of MM is undergoing a rising trend globally, with higher socio-economic status and an aging society being potential risk factors [29]. According to the Global Cancer Observatory, the age-standardized incidence rate of MM was 1.8 cases per 100,000 persons world-wide in 2022, while 201,903 new cases of MM are estimated world-wide from 2022 to 2025. The change of new cases from 2022 to 2025 represents an increase of 7.4% [30].

Regarding triple-class refractory disease stages (i.e., refractory to IMiDs, PIs, and anti-CD38 monoclonal antibodies), the median overall survival declines to approximately four to six months [31]. Complicated cases are represented by heavily pretreated patients with aggressive, relapsed, or refractory disease stages, who are often in frail, aged, and multimorbid general health conditions.

Despite previous considerable advances, the treatment of MM remains a challenge. Meanwhile, researchers have scrutinized the TME, aiming to improve the immunotherapeutic approaches and broaden the molecular–biological understanding. The TME features a complex network of distinct cellular (e.g., stromal cells, endothelial cells, stem cells, immune cells) and non-cellular elements (e.g., growth factors, cytokines, chemokines, extracellular matrix components), playing an integral part in cancer pathobiology and anti-cancer immune responses [32,33,34,35,36]. Therefore, targeting the TME would provide new approaches to disrupt the myeloma-supportive micromilieu. With regard to the growing research field of immune-oncology as well as the continuous development of cancer immunotherapy, the present review attempted to structure cancer cell biological concepts of the TME in MM from an immunological perspective. Several actors across the immune system are crucial for the TME in MM, contributing to the malignant evolution of the myeloma precursor stages and driving relapse as well as therapy resistance [37,38,39,40]. While the immunological TME is utilized by myeloma cells to realize immune escape and cancer progression, advanced oncotherapies with CAR T cells and therapeutic antibodies direct the patient’s own immune response against the myeloma cells [21,41,42,43,44,45]. Given the fact that these immunotherapeutic strategies have achieved promising results in the clinical setting, it is upon further fundamental research to explore the immunological power of novel cancer therapeutics in myeloma treatment and to investigate the underlying molecular–biological mechanisms of action.

Selinexor has expanded the treatment horizon in myeloma by selectively inhibiting the nuclear export receptor Exportin-1 (XPO1) and disrupting the nucleocytoplasmic flow, inducing cell cycle arrest and apoptosis in cancer cells [46]. While increasing evidence suggests that Selinexor also influences the TME in myeloma and other cancers on an immunological level [47,48], in depth investigations on Selinexor’s impact on anti-cancer immune responses are scarce in the current literature. Therefore, an aim of this review was to understand Selinexor’s potential to influence the immunological TME. Reflecting and expanding the scientific perspectives on this novel oncotherapy provides a basis for researchers to shape the future of myeloma treatment.

## 2. Immunological Channeling of the TME in Myeloma

The sophisticated networking of immune cells, ranging from the myeloid (e.g., monocytes, macrophages, myeloid-derived suppressor cells (MDSCs), dendritic cells (DCs)) to the lymphoid lineage (e.g., B-cells, natural killer (NK) cells, T cells) builds the immunological TME in myeloma [49].

Macrophages are characterized by a remarkable cellular plasticity and a versatile system of activity. They play significant roles in the physiologic homeostasis and tissue regeneration of the bone and BM. Moreover, macrophages form a heterogenic spectrum of phenotypes, where M1 (anti-tumoral phenotype) and M2 (pro-tumoral phenotype) are the most studied [50,51,52]. A growing body of research emphasizes the importance of macrophages regarding myeloma development and progression [53,54,55,56]. Macrophages were shown to promote the survival and dissemination of myeloma cells by secreting pro-inflammatory cytokines, such as S100 calcium-binding protein A9 (S100A9), interleukin (IL)-6, and tumor necrosis factor α (TNF-α) [57,58]. IL-6 exerts a suppressive effect on NK cells and stimulates tumor cell proliferation, survival, and metastasis in myeloma [59,60]. Different studies have further associated elevated levels of IL-6 and TNF-α with advanced disease stages in MM [61,62,63]. M2 polarized macrophages are also capable of promoting angiogenesis [64]. Tumor-associated macrophages (TAMs) specifically reside in the TME and are predominantly polarized toward the M2 phenotype, supporting the cancerous niche in MM [65,66]. Accumulating preclinical evidence has revealed a diverse impact of TAMs on the development and progression of MM. Inflammasome activation in TAMs and the subsequent secretion of IL-1β and IL-18 was found to promote tumor growth and osteolytic destruction in MM in vivo [67]. Furthermore, the cross-talk of TAMs with other immune cells of the TME was identified to potentially participate in myeloma cell immune escape [68]. In particular, the CD163^+^ subpopulation of TAMs has been recognized to infiltrate the BM niche, indicating poor prognosis in myeloma [69,70,71], while elevated levels of CD163^+^ macrophages were also detected in other hematologic and solid cancers [72,73,74,75]. It is noteworthy that macrophages increasingly express CD163, especially in response to inflammatory stimuli [76,77]. Activation of the signal transducer and activator of transcription 3 (STAT3)-related pathway further supports the immunosuppressive phenotype of CD163^+^ TAMs [78]. As myeloma patients suffer from prolonged and intensified therapy courses due to relapsed and refractory states, it is crucial to further investigate the immunological processes underlying treatment success or drug resistance. In this context, Mougiakakos et al. have demonstrated that the treatment with lenalidomide shifts the phenotype of TAMs, derived from myeloma patients, toward a pro-inflammatory and tumoricidal M1 direction through the IKAROS family zinc finger 1 (IKZF1)-interferon regulatory factor (IRF) 4/5 axis [79]. Several studies have further recognized TAMs to be involved in the development of drug resistance in MM and different other cancers [80,81,82,83,84]. Thus, in light of the immunological power to nurture the cancerous process, TAMs could display a promising target for therapeutic intervention in MM.

Another aspect is that myeloma cells dynamically interact with immune cells to shape the TME in a sustainable manner, thereby fostering their own survival. Beider et al. have demonstrated that myeloma cells and macrophages reciprocally promote the secretion of C-X-C motif chemokine ligand 13 (CXCL13) via transforming growth factor β (TGF-β) and Bruton’s tyrosine kinase signaling. This was shown to not only lead to CXCL13 driven myeloma growth but also to contribute to receptor activator of nuclear factor (NF) kΒ ligand (RANKL)-mediated osteoclastogenesis [85]. Myeloma cells can further promote the immunosuppressive polarization of TAMs via the secretion of IL-10 and extracellular vesicles [56,86]. Pucci and colleagues have found that myeloma-derived extracellular vesicles upregulate the expression of programmed death ligand 1 (PD-L1) and IL-6 in naive macrophages [87]. Moreover, macrophages have been shown to be activated by mitochondrial damage-associated molecular patterns (mtDAMP) originated from human myeloma cells, which promoted myeloma progression via the activation of the stimulator of interferon genes (STING) pathway in vivo [88]. *STING* expression has been shown to be downregulated in BM specimens of myeloma patients. Moreover, intracellular STING was reduced in neutrophils of MM patients. Interestingly, intracellular STING expression in the macrophages and NK cells was comparable to normal tissue [89].

MDSCs are suggested in a growing body of literature as being a critical component of the immune micromilieu. In this context, MDSCs have a reciprocal relationship with tumor cells of various cancers and are associated with relapse and therapy resistance [90,91,92,93,94]. Apart from that, myeloma cells are capable of inducting and homing MDSCs via paracrine mechanisms ensuring an immunosuppressive milieu, which interferes with anti-tumoral immune surveillance and adversely impacts prognosis [95,96,97,98,99]. The depletion of MDSCs has even been shown to enhance the sensitivity to immune checkpoint (IC) inhibition in vivo [100]. Another recent study found that MDSCs increasingly express calgranulin A and B, interfering with CAR T cell cytotoxicity [101].

DCs build a dynamic system of potent antigen-presenting cells in all tissues orchestrating both innate and adaptive immune responses [102]. The cross-presentation of antigens by DCs induces specific cellular responses by CD8^+^ T cells [103]. Suppression or reduction of the DC population in the TME potentially impairs anti-tumoral immune responses [104,105,106]. Myeloma cells might further promote pro-inflammatory responses via IL-1β secretion by DCs as well as the generation of tolerogenic DCs [107].

NK cells realize anti-cancer immunosurveillance through their cytotoxic and suppressive activity against malignant cells [108,109,110]. Different cytokines, such as IL-2, IL-15, IL-21, and type I and III interferons, promote NK cell proliferation and survival as well as trigger their cytotoxicity [111,112,113,114]. NK cells eliminate cancer cells by direct cell–cell contact leading either to exocytosis of perforins and granzymes or to caspase-dependent apoptosis via the activation of death receptors [115,116]. However, several studies have indicated quantitative and functional alterations of NK cell subsets in the course of myelomagenesis, potentially leading to NK cell exhaustion and impaired NK cell cytotoxicity as well as reduced NK cell effector functions in MM [117,118,119,120]. Alterations in NK cell distribution and functionality have also been associated with advanced disease stages and poor patient outcomes in MM [121,122,123].

T cells act in a bidirectional manner in the TME of various cancers. Cytotoxic T cells eradicate cancer cells, while regulatory T cells (Tregs) conduct an immunosuppressive quality and enable tumor immune escape [124,125,126]. In several studies, an elevated frequency of Tregs has been observed in myeloma patients [127,128,129,130,131]. In mice, interferons secreted by myeloma cells were found to promote the expansion of Tregs as well as their immunosuppressive capacity [132,133]. Increased levels of Tregs are further associated with impaired patient outcomes in MM [128,134]. In more recent studies, it has been indicated that Tregs navigate tumor antigen presentation and immunosuppressive signaling in MM by inducing the production of TGF-β1, downregulating the gene expression of members of the class I major histocompatibility complex (MHC) molecules, and increasing *PD-L1* expression in myeloma cells [135]. Furthermore, the depletion of Tregs enabled CD8^+^ T cell and NK cell immune response against myeloma cells [136]. In fact, Takahashi et al. have described profound myeloma control after depleting Tregs from the mobilized stem cell graft of myeloma patients [137]. It was also found that the distribution of other T cell subsets, such as IL-17-producing CD4^+^ T cells, was relevantly altered in myeloma patients and potentially influences the disease course [138,139,140]. Additionally, in different studies, a varying functional composition was proposed of the CD8^+^ T cell in myeloma, especially with regard to their differentiation and effector functions, as well as activity and exhaustion states [141,142,143]. Interestingly, the regulatory and effector T cell profiles were identified to impact the immunotherapeutic response dimension in MM as well [144,145,146,147,148].

## 3. Selinexor’s Molecular–Biological Positioning in the Treatment of Myeloma

Transportation of macromolecules across the double-membraned eukaryotic nuclear envelope is established by specific nuclear transport receptors through the nuclear pore complex. The latter is a membrane-embedded transport protein channel, composed of nucleoporin copies. Nuclear pore complexes are responsible for the homeostatic transfer of molecular cargoes between the nucleus and the cytoplasm, overall regulating the genetic flow from transcription to translation. Import and export through the nuclear pore complex is promoted by the Ras-related nuclear protein (RAN) GTPase, providing a gradient across the nuclear membrane and energy for the subsequent transport [149,150,151]. XPO1, also known as chromosome region maintenance 1, is a member of the karyopherin-β family of proteins and the main soluble nuclear export receptor in eukaryotic cells, shuttling approximately 200 signal-transducing proteins through the nuclear pore complex. Leucin-rich nuclear export signals are recognized by XPO1 to traffic molecular cargoes from the nucleus to the cytoplasm, such as tumor suppressor proteins (TSPs), cell cycle regulators, and immune response regulators [152,153]. While the balanced nuclear export is inevitable for the physiologic function of healthy cells, various hematologic and solid cancers benefit from the overexpression of XPO1 [154,155]. Therefore, the inhibition of XPO1 has offered an innovative strategy to approach antineoplastic targeting.

First achieved with a natural low-molecular-weight metabolite named Leptomycin B, which was originated from *Streptomyces*, the inhibition of XPO1 was recognized to induce cell death [156,157]. Subsequently, researchers designed the new generation of small-molecule inhibitors, the SINE compounds, which specifically target XPO1 by covalently binding to the reactive site at cysteine 528 in the XPO1 cargo-binding pocket [158]. Selinexor (marketed as XPOVIO (USA) or NEXPOVIO; formerly labeled as KPT-330) has the chemical formula C_17_H_11_F_6_N_7_O and was developed by Karyopharm Therapeutics as the first-in-class orally bioavailable SINE compound. Demonstrating a broad preclinical cytotoxic and antineoplastic activity, Selinexor paved the way for renewing myeloma treatment [159,160].

In July 2019, the US Food and Drug Administration (FDA) granted Selinexor accelerated approval in combination with dexamethasone based on the efficacy and safety results from the STORM trial (NCT02336815). Since then, Selinexor has been indicated for adult patients with relapsed or refractory MM who had received at least four prior therapies and whose disease was refractory to at least two PIs, at least two immunomodulatory agents, and one anti-CD38 monoclonal antibody [161,162]. Since December 2020, Selinexor has been further FDA approved in combination with bortezomib and dexamethasone for the treatment of adult patients with MM who had received at least one prior therapy, according to the evaluated efficacy by the BOSTON trial (NCT03110562) [27,162]. In addition, clinical findings from the STOMP (NCT02343042) and the NCT02199665 trials have demonstrated the efficacy of Selinexor and dexamethasone in combination with bortezomib or carfilzomib in patients with MM refractory to PIs [163,164,165,166]. Regarding the actual state of research, Selinexor’s molecular–biological action involves several mechanisms focusing the nucleocytoplasmic flow of cancer-related cargoes, deoxyribonucleic acid (DNA) damage repair, glucocorticoid signaling, and osteo-metabolism (Figure 2).

Selinexor particularly retains TSPs (e.g., p53, retinoblastoma (Rb), p21, p27, adenomatous polyposis coli (APC), forkhead box O (FOXO)) in the nucleus leading to, inter alia, rapid cell cycle arrest and apoptosis of the cancer cells [167,168,169,170,171,172,173,174,175]. Other cancer-related XPO1 cargoes are related to cell cycle regulators (e.g., cyclin dependent kinase 1 (CDK1), cyclin B1/D1), cell survival proteins (e.g., survivin, cellular inhibitor of apoptosis protein 1 (cIAP1), myeloid cell leukemia 1 (MCL1), insulin-like growth factor binding protein 2 (IGFBP2), telomerase reverse transcriptase (TERT)), and autophagy-related proteins (e.g., beclin 1, centrin, yes-associated protein 1 (YAP1)). Different growth regulators (e.g., KIT, epidermal growth factor receptor (EGFR), FMS related receptor tyrosine kinase 3 (FLT3), B-Raf proto-oncogene, phosphatase and tensin homolog (PTEN), phosphatidylinositol 3-kinase (PI3K), protein kinase B/Akt, Ras protein specific guanine nucleotide releasing factor 1 (Ras-GRF1), and ABL proto-oncogene 1), members of the IRF family (IRF3, IRF5), cytosolic steroid receptors (peroxisome proliferator-activated receptor γ (PPAR-γ)), mediators of cell-signaling transduction pathways (NF-kB inhibitor α), epithelial-mesenchymal transition proteins (Snail), as well as key drug targets (Topoisomerase IIα) are also shuttled via XPO1 and display key roles in cancer pathogenesis [167,168,176,177,178,179,180,181]. The deregulation of cell biological signaling, notably the p53 and the NF-kB pathways, is a hallmark of myeloma development and progression [195,196,197,198]. While a large number of cancer-related cargoes of XPO1 are already recognized, detailed investigations on Selinexor interference with the specific signaling pathways are limited.

Selinexor also traps the messenger ribonucleic acid (mRNA) cap-binding protein eukaryotic translation initiation factor 4E (eIF4E) in the nucleus, which is responsible for the nuclear export of mRNAs encoding proto-oncogenes. In this way, Selinexor is capable of reducing the ribosomal translation of proto-oncogenes including the MYC oncogenic program, the apoptosis regulators B-cell lymphoma (BCL) 2 and 6, as well as the molecular chaperone heat shock protein 70 (HSp70) [182,183,184,185,186]. Moreover, Selinexor impacts the DNA damage repair of cancer cells, which is fundamental for the high cellular turnover during malignant progression. Interestingly, XPO1 overexpression in cancer cells increases the tolerance to genotoxic stress due to the enhanced nuclear export of eIF4E carrying DNA damage repair mRNAs [187]. In this context, Selinexor reduces the expression of DNA damage repair proteins on a transcriptional and translational level, potentially overcoming the resistance to DNA damage-inducing therapies [188,189]. These findings also suggest that Selinexor could interlink different antineoplastic approaches in myeloma treatment and possibly circumvent therapy resistance.

Dexamethasone is part of the first-line regimen of MM treatment and is also included in the therapy of advanced myeloma stages. Combining Selinexor with dexamethasone has improved the overall survival rates of intensively treated MM patients in quad- and penta-refractory stages [190]. It has been demonstrated that Selinexor induces the expression of the glucocorticoid receptor and increases the transcriptional activity of the glucocorticoid receptor in combination with dexamethasone. Selinexor was found to inhibit the mammalian target of rapamycin (mTOR) pathway in synergy with dexamethasone, thereby promoting cell death in myeloma cells. In fact, the nuclear accumulation of the glucocorticoid receptor was shown to enhance the expression of *REDD1*, which is significant for the inhibition of the mTOR complex 1 [191,192].

Myeloma-induced osteodestruction and consecutive pathologic fractures dramatically impair patient morbidity and mortality. In these cases, a multimodal and interdisciplinary management, including osteoprotective medication with bisphosphonates or the RANKL inhibitor denosumab, as well as local irradiation and orthopedic intervention, is required [12]. Therefore, patients would benefit from the integration of osteoprotective antineoplastic substances. While the literature contains hints on Selinexor’s influence on osteolysis, the detailed molecular–biological mechanisms underlying its mode of action are yet to be understood completely. In 2014, Tai and colleagues demonstrated that SINEs block the NF-kB activity in myeloma cells potentially by trapping the NF-kB inhibitor α in the nucleus. Moreover, SINEs were found to directly block the RANKL-induced NF-kB/p65 activity and the induction of NF of activated T cells cytoplasmic 1 (NFATc1) in osteoclast precursors [169]. Regarding other tumor entities, Selinexor was observed to impair the secretion of pro-osteolytic cytokines and reduce osteoclastogenesis in prostate cancer cells in vitro. Osteolytic lesions and serum levels of osteoclast markers were seen to be further reduced by Selinexor in the metastatic prostate carcinoma model in vivo [193]. Recently, Chen and co-workers showed that Selinexor inhibits RANKL-induced osteoclast formation from BM-derived macrophages in vitro [194]. In light of these findings, Selinexor displays the potential to impact and possibly shape key cellular pathways of the osteo-metabolism, which might provide additional benefit to the cancer treatment, especially with regard to bone tissue involvement due to primary cancer manifestation or metastasis.

## 4. Selinexor’s Impact on the Immunological TME

How Selinexor might influence immune cells of the TME is a question that researchers are seeking to answer. The preclinical data and clinical findings already contain several hints on Selinexor’s capacity to impact different immune cells of the myeloid and lymphoid lineage (Figure 3).

Macrophages are considered to be relevant actors in the immunological micromilieu of cancer and inflammation [215,216,217,218]. Therefore, it is all the more important to precisely evaluate novel immunotherapeutics regarding their impact on the TME. According to Jiménez et al., Selinexor-containing treatment is able to shift the polarization of TAMs toward a pro-inflammatory M1 polarized phenotype. Selinexor-containing treatment also decreased the expression of programmed death 1 (PD-1) and signal regulatory protein α (SIRPα) in the remaining M2 polarized TAMs in vivo [199]. In a recently published mouse model with subcutaneously injected B-cell lymphoma cells, Selinexor administration reduced the amount of M2 polarized macrophages in the tumor specimens [200]. Selinexor’s influence on TAMs might potentially redirect the TME toward a pro-inflammatory milieu, which could support anti-cancer immune responses and promote the effect of other immunotherapies.

Regarding MDSCs, the IL-6/STAT3 pathway was recently identified to upregulate the expression of *XPO1*, resulting in the activation of the extracellular-signal regulated kinases (ERKs)-mediated mitogen-activated protein kinase (MAPK) pathway. STAT3-mediated *XPO1* expression potentially supports MDSC fitness and duration in the TME. The blocking of XPO1 led to the transformation of MDSCs into immunostimulatory neutrophil-like cells as ERK1/2 accumulated in the nucleus and decreased the transcriptional activity of MDSCs [201].

NK cells recognize and destroy cancer cells by receptor-mediated signaling. Selinexor was shown to downregulate the surface expression of human leukocyte antigen (HLA)-E on lymphoma cell lines and on primary chronic lymphocytic leukemia cells. As HLA-E is a ligand for the inhibitory surface receptor CD159a on NK cells, Selinexor was thus suggested to increase the anti-cancer immune response of NK cells in vitro [202,203]. In addition, Hu et al. analyzed the gene expression data of myeloma patients and identified that the expression of *ABCC4* is positively correlated with NK cell infiltration and associated with the sensitivity to Selinexor treatment [219].

Modulating anti-cancer immune responses is crucial for modern immune-oncotherapy. As part of recent investigations, the impact of XPO1 inhibition by Selinexor has been discussed on T cell activity, functionality, and homeostasis, especially with regard to combating the phenomenon of T cell exhaustion. In this context, Tyler et al. described a dose-dependent effect of Selinexor on T cell development and function. They found that increased dose and application frequency of Selinexor transiently impaired murine CD8^+^ T cell activity in vitro. The authors further investigated CD8^+^ T cell fitness and effector functions in a syngeneic mouse model of melanoma. Treatment with clinically relevant doses and schedule intervals of Selinexor caused there to be enhanced CD8^+^ T cell fitness, reduced IC expression (*PD-1*, *TIM1*, *CD44*), and preserved T cell effector functions [204]. Farren et al. demonstrated that Selinexor induces the gene expression of *PD-1* and *CTLA4* in leucocytes and human melanoma cell lines in vitro. Moreover, the authors found that IC inhibitors combined with Selinexor significantly reduced melanoma tumor burden in vivo [220]. In the context of CAR T cell therapy, the pretreatment with SINE compounds improved the cytotoxic potential of CD19-directed CAR T cells against Burkitt’s lymphoma and acute lymphocytic leukemia cell lines in vitro [206]. In a mouse model of non-Hodgkin lymphoma, Stadel et al. observed that the administration of Selinexor prior to CAR T cells significantly reduced tumor burden compared to CAR T cell therapy in non-Selinexor-treated mice or Selinexor monotherapy. The results of this study suggest that Selinexor might sensitize cancer cells to CAR T cell mediated cytotoxicity [214], while the detailed underlying mechanisms still need to be elucidated. As part of another in vitro study on human T cells, it was shown that increasing doses of Selinexor inhibited proliferation, promoted apoptosis, and increased the differentiation into Tregs. However, cytokine secretion (IL-2, TNF-α, IF-γ, IL-4, IL-6, IL-10) and gene expression of exhaustion markers (*PD-1*, *TIM3*, *LAG3*) were not affected in the T cells by Selinexor [205].

As far as can be understood from existing clinical data, Selinexor might contribute to an immunomodulating effect regarding the response behavior to targeted immunotherapies. For example, the pretreatment with Selinexor improved the response of patients with triple-negative breast cancer to subsequent bispecific antibody therapy [207]. In myeloma, Selinexor-containing treatment displayed potency with durable responses and treatment tolerability in heavily pretreated patients, even those refractory to antibody drug conjugate anti-BCMA therapy [208]. In addition, Selinexor did not negatively impact the overall survival of MM patients, who received subsequent non-cellular anti-BCMA therapy [209]. In another work, promising data were shown on the combination of Selinexor with PD-1 blockade in advanced and refractory patients with NK/T cell lymphoma resistant to prior anti-PD-1 antibody treatment [210]. Moreover, Chari et al. have indicated that extensively pretreated patients, with rapidly progressing myeloma even after BCMA directed CAR T cell therapy, potentially benefit from the combination of Selinexor with low-dose dexamethasone, with or without PIs (bortezomib or carfilzomib) [211]. In another clinical investigation, it has been reported that Selinexor-containing salvage therapy improved the survival of patients who relapsed after BCMA-directed CAR T cell therapy [212]. Wang et al. presented preliminary data of two patients with extramedullary MM receiving Selinexor as a bridging therapy prior to CAR T cells. Both patients achieved stringent complete remission with survival over 13 and 10 months, respectively [213].

## 5. Conclusions

Selinexor, the first-in-class orally bioavailable SINE compound, is an innovative agent in the repertoire for MM treatment and displays the potential to influence the immunological TME in myeloma. The results of recent research efforts have offered insights into the complex interplay of immune cells during MM initiation, progression, and therapy resistance, as well as relapse. However, the current state of knowledge highlights the present need to explore the impact of the new generation of cancer therapeutics on the immune-oncological axis in depth. Novel understanding in this area would enable the evaluation of Selinexor’s immunomodulatory route and further provide fresh approaches to the expanding research field of immune-oncotherapy. Investigating target points of cancer-related signaling pathways, aberrant in myeloma and immune cells, could not only improve present antineoplastic strategies but also identify potential bypassing concepts to combat therapy resistance. Selinexor’s role in the advanced immunotherapeutic landscape, as provided by therapeutic antibodies and CAR T cells, is expected to shape the future directions of immune-oncotherapy in MM. Finally, further in vitro and in vivo experimental studies with primary human cells and randomized controlled clinical trials on Selinexor sequencing regimens are needed to expand the horizons of knowledge in this area.

## Figures and Tables

**Figure 1 cells-14-00430-f001:**
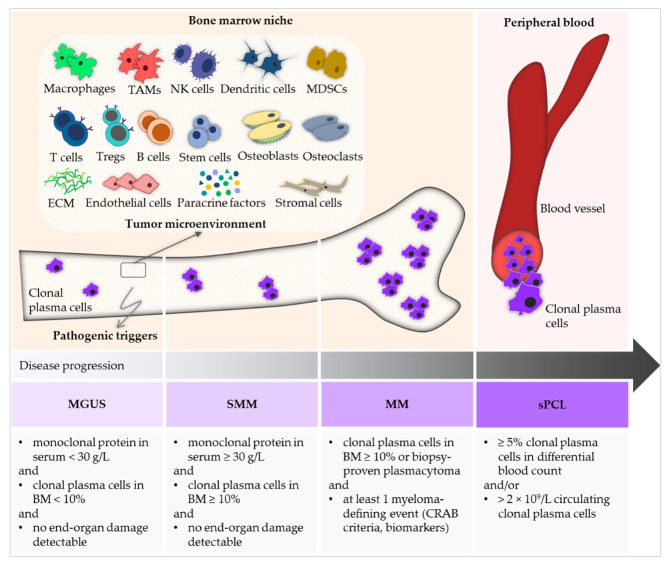
Schematic illustration of the development and diagnosis of malignant plasma cell dyscrasias. Monoclonal gammopathy of undetermined significance (MGUS) and smoldering multiple myeloma (SMM) represent the precursor states of multiple myeloma (MM). Secondary plasma cell leukemia (sPCL) displays the dissemination of clonal plasma cells into the blood following previously diagnosed MM. Disease progression is influenced by, inter alia, various pathogenic triggers as well as the tumor microenvironment in the bone marrow niche. The diagnostic criteria of MGUS, SMM, MM, and sPCL are summarized according to the International Myeloma Working Group updated criteria [1]. ECM, extracellular matrix; MDSCs, myeloid-derived suppressor cells; NK, natural killer; TAMs, tumor-associated macrophages; Tregs, regulatory T cells.

**Figure 2 cells-14-00430-f002:**
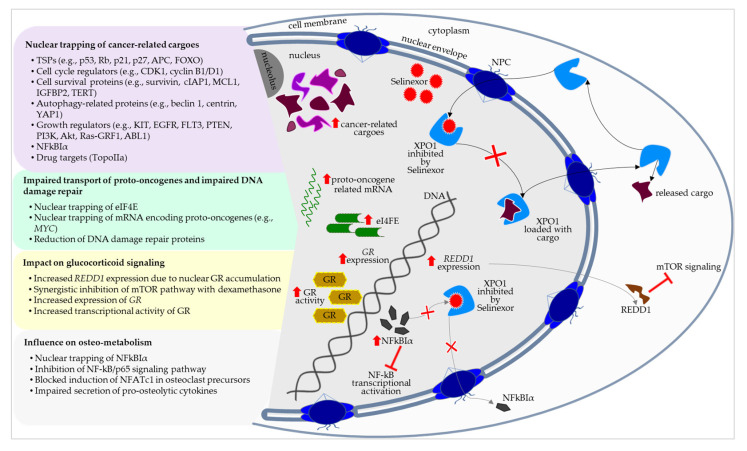
Schematic illustration of Selinexor’s mode of action in myeloma cells. Selinexor inhibits Exportin-1 (XPO1) and interrupts the nucleocytoplasmic flow through the nuclear pore complex (NPC). Cancer-related cargoes and other nuclear export signaling molecules are trapped in the nucleus [167,168,169,170,171,172,173,174,175,176,177,178,179,180,181], influencing deoxyribonucleic acid (DNA) damage repair [182,183,184,185,186,187,188,189], glucocorticoid signaling [190,191,192], and osteo-metabolism [169,193,194]. Red arrowheads indicate inhibition. Red upright flashes indicate upregulation. Red crosses indicate interruption. ABL1, ABL proto-oncogene 1; Akt, protein kinase B; APC, adenomatous polyposis coli; CDK1, cyclin dependent kinase 1; cIAP1, cellular inhibitor of apoptosis protein 1; EGFR, epidermal growth factor receptor; eIF4E, eukaryotic translation initiation factor 4E; FLT3, FMS related receptor tyrosine kinase 3; FOXO, forkhead box O; GR, glucocorticoid receptor; IGFBP2, insulin-like growth factor binding protein 2; MCL1, myeloid cell leukemia 1; mRNA, messenger ribonucleic acid; mTOR, mammalian target of rapamycin; NFATc1, nuclear factor of activated T cells cytoplasmic 1; NFkBIα, nuclear factor kB inhibitor α; PI3K, phosphatidylinositol 3-kinase; PPAR-γ, peroxisome proliferator-activated receptor γ; PTEN, phosphatase and tensin homolog; RANKL, receptor activator of nuclear factor kΒ ligand; Ras-GRF1, Ras protein specific guanine nucleotide releasing factor 1; Rb, retinoblastoma protein; REDD1, regulated in development and DNA damage responses 1; TERT, telomerase reverse transcriptase; TopoIIα, Topoisomerase IIα; TSP, tumor suppressor protein; YAP1, yes-associated protein 1.

**Figure 3 cells-14-00430-f003:**
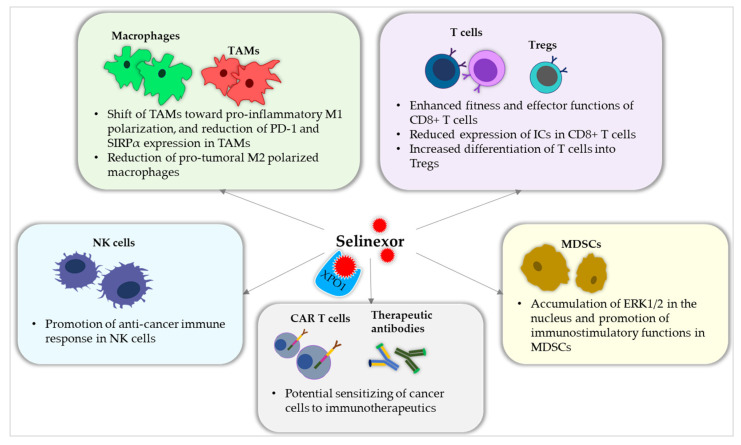
Schematic illustration of Selinexor’s influences on immune cells and immunotherapy. Selinexor is suggested to impact macrophages and tumor-associated macrophages (TAMs) [199,200], myeloid-derived suppressor cells (MDSCs) [201], natural killer (NK) cells [202,203], and T cells [204,205] in the tumor microenvironment. Selinexor potentially sensitizes cancer cells to chimeric antigen receptor (CAR) T cells and therapeutic antibodies [206,207,208,209,210,211,212,213,214]. CD8, cluster of differentiation 8; ERK1/2, extracellular-signal regulated kinases 1/2; IC, immune checkpoint; PD-1, programmed death 1; SIRPα, signal regulatory protein α; Tregs, regulatory T cells.

## Data Availability

No new data were created or analyzed in this study.

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
