# Peer review of "Selinexor’s Immunomodulatory Impact in Advancing Multiple Myeloma Treatment"

_cells, 2025, doi:10.3390/cells14060430_

Round 1
Reviewer 1 Report
Comments and Suggestions for Authors
SUMMARY: This is a well-written and comprehensive review article outlining the impact of Selinexor on the tumor microenvironment in multiple myeloma. Suggestions to improve the manuscript are outlined below:
Major Comments:
- The Introduction could be improved by including a schematic figure summarizing the different stages of multiple myeloma and how they differ from one another.
- Line 223: The entire section would benefit from a strong concluding sentence that wraps up the material presented.
- Why did the authors focus on only the immunological tumor microenvironment? What about the stromal cells, epithelial cells, endothelial cells, osteoblasts, and osteoclasts? Are they impacted by Selinexor treatment?
Minor Comments:
- Lines 70-72: 2022 data is old, please update with more current numbers for MM incidence.
- The number of abbreviations makes the review article more difficult to read.
- Figures 1 and 2 could be improved by including bullet points in front of the different statements.
Reviewer 2 Report
Comments and Suggestions for Authors
The paper is potentially interesting
However, corrections are needed to make the paper interesting
1. I think some parts should be reordered
2. After the introductory part about MM, and the basic characteristics, my opinion is that the immunological disorders should first be written about, which are here marked as a chapter in the paper marked under number 3. Immunological Channeling of the TME in Myeloma.
3. in that part of the paper, be sure to include references related to immune disorders in myeloma during disease progression under the title: "Clinical stage-dependent decrease of NK cell activity in multiple myeloma patients" which correlate with other important clinical parameters necessary for diagnosis and explanation of the mechanism of the disorder
4. Based on indicators of impaired immune system function in tumor elimination and general immunosuppression, which should be added to the literature: "Decreased CD161 activating and increased CD158a inhibitory receptor expression on NK cells underlies impaired NK cell cytotoxicity in patients with multiple myeloma," new forms of immunomodulatory therapy based on receptor blockade were later suggested.
5. There is also a disorder in other cell populations in myeloma and plasmacytoma "during the progression of the disease, which has been shown in the literature and refers to IL-2 production, the number of T cells in peripheral blood and other subpopulations:
6. I think that this should be discussed first and only then should chapter number 2, which refers to therapy, be called 2. Selinexor's Molecular-biological Positioning in the Treatment of Myeloma.
7. In the part of the text where the role of TNF and IL-6 in the pathophysiological mechanisms associated with MM is mentioned, an important reference should certainly be added regarding the cytokine TNF, which has been published in the literature and which discusses elevated values ​​of proinflammatory mediators TNF, called "Correlation of sera TNF-alpha with percentage of bone marrow plasma cells, LDH, beta2-microglobulin, and clinical stage in multiple myeloma"
Comments on the Quality of English Language
corection
Round 2
Reviewer 1 Report
Comments and Suggestions for Authors
The authors have adequately addressed my concerns.
Reviewer 2 Report
Comments and Suggestions for Authors
Acept this version
Comments on the Quality of English Language
Ok